# Stem Cell Therapy for Sequestration of Traumatic Brain Injury-Induced Inflammation

**DOI:** 10.3390/ijms231810286

**Published:** 2022-09-07

**Authors:** Mia C. Borlongan, Susanna Rosi

**Affiliations:** 1Department of Physical Therapy and Rehabilitation Science, University of California, San Francisco, CA 94143, USA; 2Brain and Spinal Injury Center, University of California, San Francisco, CA 94110, USA; 3Department of Neurological Surgery, University of California, San Francisco, CA 94143, USA; 4Weill Institute for Neuroscience, University of California, San Francisco, CA 94158, USA

**Keywords:** trauma, traumatic brain injury, stem cell-based therapy, resident stem cells, neurogenesis, inflammation, brain repair

## Abstract

Traumatic brain injury (TBI) is one of the leading causes of long-term neurological disabilities in the world. TBI is a signature disease for soldiers and veterans, but also affects civilians, including adults and children. Following TBI, the brain resident and immune cells turn into a “reactive” state, characterized by the production of inflammatory mediators that contribute to the development of cognitive deficits. Other injuries to the brain, including radiation exposure, may trigger TBI-like pathology, characterized by inflammation. Currently there are no treatments to prevent or reverse the deleterious consequences of brain trauma. The recognition that TBI predisposes stem cell alterations suggests that stem cell-based therapies stand as a potential treatment for TBI. Here, we discuss the inflamed brain after TBI and radiation injury. We further review the status of stem cells in the inflamed brain and the applications of cell therapy in sequestering inflammation in TBI.

## 1. Traumatic Brain Injury and Inflammation

Traumatic brain injury (TBI) is a growing health problem due to the chronic behavioral and cognitive impairments that affect the quality of life of millions of individuals [1,2,3,4,5]. Its incidence is between 1.5–3.8 million people, who succumb to TBI each year in the United States alone [6,7]. In terms of age-specific TBI incidence, Americans aged 0 to 4 years show the highest rate of TBI-related emergency department visits (about 1200 per 100,000 population), then those aged 15 to 19 years (about 750 per 100,000), and the most prone to hospitalizations (about 340 per 100,000) and deaths (about 60 per 100,000) are those aged 75 years and older [8,9,10,11]. For sex-specific TBI incidence, men appear to suffer TBI about 40% more than women in the general adult population, but this sex difference diminishes at age 75 years and older [12,13]. Mortality rate reveals that TBI accounts for about 30% of all injury-related deaths in the United States [12,13]. Severity classification of TBI has largely categorized the injury into three types, namely, mild, moderate, and severe, based on the normal, partially abnormal, and fully abnormal structural appearance of the skull, respectively, with corresponding loss of consciousness at less than 30 min, between 30 min and 24 h, and more than 24 h following the initial injurious episode [14]. TBI prognosis is challenging because of disease heterogeneity, but Glasgow Coma Scale motor score, age, and pupillary activity appear to serve as good prognostic factors [15]. Similarly, TBI treatments vary, with each case approached differently, based on the severity of the injury, which may include non-invasive modalities, such as drug administration, and invasive surgery, including bilateral decompressive craniectomies [16]. Moreover, while TBI has been traditionally considered to be an acute injury, survivors manifest many pathological features and symptoms reminiscent of neurodegenerative disorders, most notable Alzheimer’s disease. Indeed, there is an increased incidence of Alzheimer-like dementia after TBI, especially in aged patients [17,18]. Although the common definition of TBI entails a mechanical injury to the skull and brain tissue, a mild form of TBI that manifests without visible physical deformation of the skull has been well documented. Indeed, mild TBI is much more rampant in our soldiers and veterans [19,20,21]. Long-term symptoms of mild TBI may include chronic traumatic encephalopathy [22], Alzheimer’s disease [23], post-traumatic stress syndrome [24], and a realm of many symptoms of neurodegenerative disorders [25]. Multiple cell death mechanisms have been implicated in TBI, but a common pathologic manifestation involves aberrant inflammation [26,27,28]. Imaging and cytokine plasma, or cerebrospinal fluid (CSF) profiling, in both mild TBI patients [29,30,31] and animal models [16,17,18] reveal upregulated inflammation. The widely accepted TBI biomarkers employ inflammation-based assays, including glial fibrillary acidic protein, ubiquitin C-terminal hydrolase-L1, and C-reactive protein [24,25,26,27,28,29,30,31,32,33,34]. Reactive microglia and infiltration of peripherally derived macrophages accompany the disease progression of TBI, especially in the aged brain [35,36,37,38,39]. More specifically, our group demonstrated that peripherally-derived macrophages (CCR2^+^) propagate to the injured brain and participate in chronic TBI-induced cognitive deficits in young animals [35,38]. Using CX3CR1(GFP/+)CCR2(RFP/+) reporter mice, we detected that TBI triggered an increase in peripherally derived CCR2(+) macrophages within the hippocampus, a well-established brain area responsible for learning and memory [35]. Moreover, we dissected the key contribution of CCR2(+) macrophages to the resulting neuroinflammation in response to TBI [35]. Finally, we showed that targeting CCR2(+) macrophages with CCX872, a robust CCR2 selective antagonist, dampened the aberrant proliferation of inflammatory macrophages after TBI [35]. Altogether, we advanced a macrophage-directed therapy that effectively blocked CCR2 and attenuated TBI-induced inflammation, coincident with amelioration of hippocampus-associated cognitive deficits [35]. Importantly, in the aged brain we measured a remarkably higher number of peripherally-derived macrophages (i.e., monocytes) after TBI than in young, injured animals [38]. In our subsequent study, we examined the effects of aging on the crosstalk between sub-chronic response to TBI of peripherally-derived monocytes (CD45^hi^; CCR2^+^) and the onset of chronic cognitive deficits. We found that TBI aged mice displayed more upregulation of peripherally-derived monocytes than the TBI young animals [38]. Such elevated peripherally-derived monocytes positively correlated with enhanced CCR2 expression [38]. In tandem, the TBI aged animals exhibited more dysregulated myeloid cell populations coupled with deficient anti-inflammatory responses than the TBI young animals. In the circulation, blood CCR2^+^ monocyte population was increased in TBI aged animals but not in TBI young animals. Lastly, genetic depletion of CCR2 suppressed the infiltration of peripherally-derived monocytes and blocked chronic TBI-induced spatial memory deficits in TBI aged animals [38]. These two studies highlight that the surge of CCR2^+^ peripherally-derived macrophages is coupled with increased levels of CCL2 chemotactic ligands and upregulated blood CCR2^+^ monocyte population, likely contributing to the impaired inflammatory responses that plagued the TBI aged animals much more than the TBI young animals [22,25]. Interestingly, the aberrant infiltration of CCR2^+^ peripherally-derived macrophages measured up to seven days post TBI in aged animals could be sequestered by knocking out CCR2 leading to attenuation of spatial memory deficits [35,38]. These findings implicate the critical involvement of resident microglia and peripherally derived macrophages in TBI pathology and treatment [35,38].

The cellular and molecular changes that lead to chronic cognitive deficits in response to inflammation are associated, at least in part, with the complement initiation components C1q, C3, and CR3, which are known to regulate microglia-synapse interactions [39]. Both genetic and pharmacological blockade of the complement pathway prevented memory deficits in aged, injured animals [39]. Altogether, these preclinical studies represent some of the compelling preclinical evidence supporting the key role of inflammation in TBI-induced long term cognitive deficits in rodents.

## 2. Inflammation-Plagued TBI-like Events

Recognizing that TBI manifests an aberrant inflammation provides the impetus to consider subtle events that at first glance appear harmless but which can result in a detrimental condition reminiscent of TBI. Accumulating evidence points to uncontrolled radiation exposure arising from radiological terrorism, industrial accidents or military circumstances poses a serious threat for civilians [40,41,42]. Interestingly, irradiation increased the vulnerability of animals to exhibit cognitive deficits when subsequently exposed to TBI [40]. Along this line, although irradiation affords therapeutic effects against brain tumors, this treatment also induces long-term persistent cognitive impairments that are dependent on inflammatory cells [41,42,43]. Similar to what is observed in rodents after TBI [35] CCR2 deficiency prevents neuronal dysfunction and cognitive deficits after exposure to therapeutic brain irradiation [31] and radiation and TBI combined injury [40,44].

Moreover, the use of whole-body irradiation to facilitate bone marrow engraftment leads to long-term and brain-wide cognitive and synaptic deficits [45]. Indeed, cognizant of the gamut of stressors that accompany deep space travels, (including ionizing radiation, gravitational changes during flight and in orbit, psychological stress from a confined environment and social isolation) galactic cosmic radiation exposure represents the most dangerous of stressors [46,47]. Each deep spaceflight stressor, or a combination thereof, may represent unique circulating immune responses coupled with an altered immune system, as evidenced by circulating plasma microRNA sequence analysis [48]. Space radiation-induced dysregulated brain inflammation is also reflected in the spleen [40,49], suggesting a robust and whole-body alteration in immune response, which resembles the TBI signature of brain and splenic inflammation [50,51,52]. While still scarce, these reports suggest that radiation-exposed individuals (from uncontrolled sources, therapeutic intervention and space exploration) may be vulnerable to upregulated central and peripheral inflammation, warranting their careful monitoring for TBI-like symptoms and treatment.

## 3. TBI-Induced Stem Cell Dysfunction: Potential of Stem Cell-Based Therapies

### 3.1. The Bone Marrow Niche and Hematopoiesis

The bone marrow niche consists of multiple cell types, including hematopoietic stem cells, interacting with extracellular matrix, chemical, and physical factors, altogether forming the microenvironment responsible for maintaining blood cell formation (hematopoiesis) from development to adulthood [53,54]. In healthy normal development and maturation, the bone marrow niche regulates the fate and function of hematopoietic stem cells and their progeny primarily contributing to healthy aging [55]. However, during injury, such as TBI, dysregulation of the bone marrow niche may occur leading to the loss of its pro-survival function [56]. Under this pathologic setting, the bone marrow niche homeostasis is perturbed, switching to a pro-death machinery (Figure 1). Indeed, the injury-induced dysregulation of the bone marrow niche is associated with an inflammatory milieu [57,58]. These documented divergent pro-survival and pro-death functions suggest the double-edged sword role of the bone marrow niche in both homeostatic and inflammatory processes in normal and pathologic conditions, respectively. Notwithstanding, the bone marrow niche has been shown to be involved in the maintenance and dysregulation of hematopoietic stem cells [59] that either facilitate cell survival or cell death. The bone marrow niche may be disrupted by TBI indirectly by signaling molecules in the circulation, suggesting crosstalk between the bone marrow and the injured brain communicating through the inflammatory and lymphatic systems [33,50,51]. Such bone marrow–brain interaction, via the circulation, may confer pro-survival and pro-death occurring, most likely, in both the bloodstream and the brain.

### 3.2. TBI and the Hematopoietic Response

Neurogenic signals are released by the injured brain, which may potentially affect the bone marrow niche or the hematopoietic response to neural injuries [60]. The process of myelopoiesis and the subsequent inflammation in response to injury may propel innate immune cells, such as neutrophils, dendritic cells and monocytes, originating from myeloid progenitor cells, to the site of injury [61,62]. Animal models of ischemic and hemorrhagic stroke detect the occurrence of injury-induced myelopoiesis [63,64]. With some overlapping pathology of brain damage between stroke and TBI, this hematopoietic response to neural injury seen in stroke may also accompany TBI. Indeed, preclinical studies in CNS trauma demonstrate similar myelopoiesis, followed by subsequent inflammation [65,66]. Accumulating evidence also implicates the participation of myelopoiesis and inflammation in space flight and irradiation [67,68]. Taken together, these findings support the notion of a crosstalk between myelopoiesis and inflammation in response to neural injury.

### 3.3. TBI-like Events and Inflammation

The inflammation-ridden pathologic feature of TBI closely approximates the onset and progressive deterioration of debilitating sensory-motor deficits and learning and memory impairments in patients [69]. A closer examination of this inflammatory signature reveals infiltration of inflammatory cells with increased pro-inflammatory cytokines in neurogenic niches in the injured brain [70,71,72]. In parallel, irradiation-induced cognitive impairments are due to disruption of hippocampal neurogenesis [73].

Space flights also alter stem cell adhesion [74], with RNA-sequencing analysis detecting specific genes differentially expressed and associated with mitochondrial metabolism [75]. These new data implicate the major involvement of stem cells in TBI and TBI-like pathology, again highlighting the close association of inflammation with the behavioral and neurostructural manifestations of the disease.

### 3.4. Stem Cell Therapy for Reducing Neural Injury-Induced Inflammation

To this end, the fact that inflammation plays an exacerbating factor in TBI acts as the impetus for finding a treatment that dampens inflammation to retard the long-term deleterious consequences of TBI. Stem cell therapy as an approach to replenish injured cells, as well as a delivery vehicle for therapeutic molecules, including anti-inflammatory factors, may serve as an appealing TBI treatment (Figure 2). In the laboratory, cell transplantation either directly into the injured brain or peripherally promotes functional recovery in TBI [76,77,78,79,80,81].

Mobilizing resident stem cells, via small molecules, such as granulocyte colony stimulating factor, also affords therapeutic effects in TBI [82,83,84,85]. As noted above, TBI-induced inflammation is rampant in both the brain and the periphery [86,87,88,89]. Accordingly, when contemplating stem cell therapy for TBI, deposition of the grafted stem cells and their secreted factors are most optimal if they hone to the brain as well as to the inflammation-enriched organs, i.e., spleen, to sequester inflammation. In the clinic, stem cell delivery via a lumbar puncture or intravenously reveals safety and modest efficacy in enhancing motor function [90,91,92,93]. Importantly, the improved clinical outcomes coincided with downregulation of inflammatory cytokines IL-1β and IFN-γ [90], which mimicked previously reported preclinical evidence in animal models of TBI [69,70] These findings support the application of stem cell therapy for dampening inflammation and improving motor functions in TBI.

## 4. Conclusions

TBI remains a significant unmet clinical need with limited therapeutic options. Inflammation, both centrally and peripherally, stands as a main pathologic feature of TBI. Mild TBI and radiation exposure results in inflammation that accompanies the onset and progression of cognitive dysfunctions coincident with stem cell dysregulation (Figure 3). Targeting inflammation-plagued organs with stem cells likely confers therapeutic effects on TBI.

## 5. Future Perspectives 

Neural injury following TBI may evolve into a progressive neurodegenerative disorder. After brain radiotherapy in humans and rodents, and after exposure to space radiation in rodents, pathology and symptoms reminiscent of TBI have been documented. The inflammation in response to brain injury is a key cell death mechanism that can exacerbate the secondary cell death of TBI. Targeting this deleterious inflammation may reduce TBI-induced secondary cell death and its associated functional impairments. To this end, the bone marrow niche stands as a potent microenvironment with a dual action that can either contribute to cell survival or cell death. Recognizing the neurogenic signals and the inflammatory cells may allow a better modulation of the bone marrow niche towards the repair of the injured brain. In particular, the bone marrow is a rich source of stem cells, which can be harvested and transplanted to TBI patients, as well as those diagnosed with similar brain injuries arising from radiation exposure. In the future, elucidating the crosstalk between the inflammatory response to TBI and the bone marrow niche, with special attention to stem cell-based therapy, may prove therapeutically beneficial in our understanding of the pathology and treatment of TBI and TBI-like events.

## Figures and Tables

**Figure 1 ijms-23-10286-f001:**
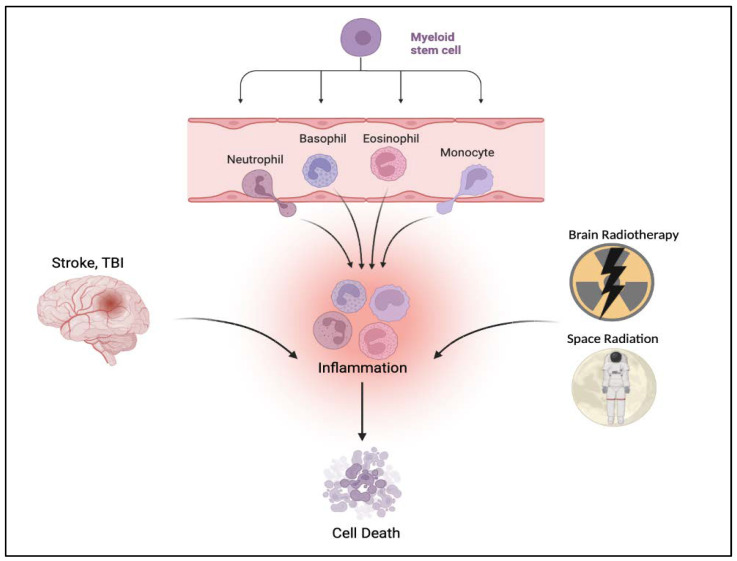
Inflammatory cells, such as neutrophils, dendritic cells and monocytes, arising from myeloid progenitor cells, infiltrate the brain altogether triggering inflammation then cell death in response to neural injury caused by stroke, TBI, brain radiotherapy and space radiation. Figure constructed via Biorender.com.

**Figure 2 ijms-23-10286-f002:**
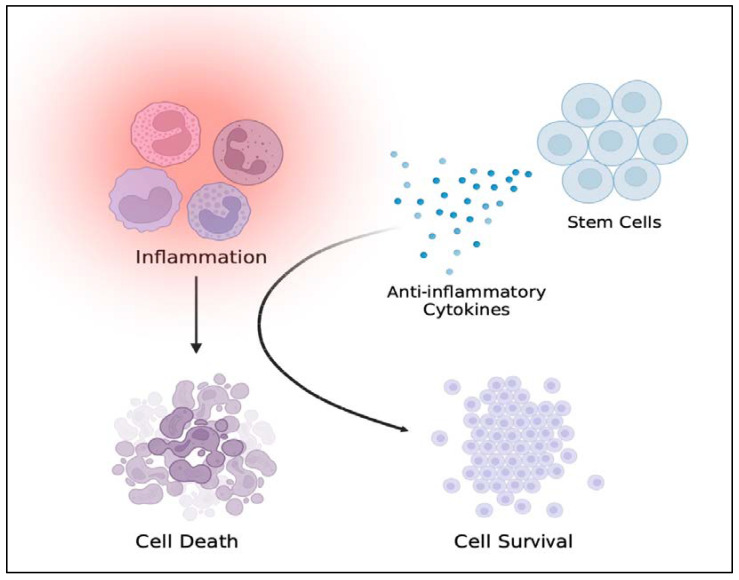
Inflammatory/immune cells are sequestered by stem cells that release anti-inflammatory factors suggesting the potential of stem cell therapy for treating neural injury. Figure constructed via Biorender.com.

**Figure 3 ijms-23-10286-f003:**
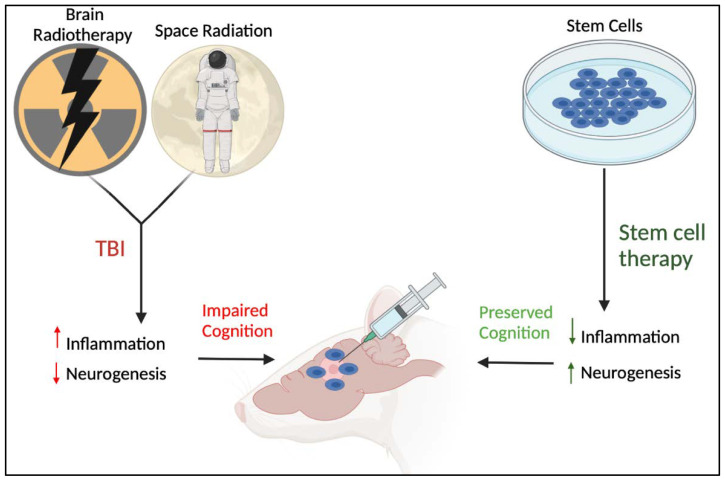
Compelling evidence suggests that brain radiotherapy and space radiation produce TBI-like pathology characterized by aberrant deleterious inflammation and dysfunctional neurogenesis resulting in impaired cognition. Stem cell-based therapies can sequester inflammation and enhance stem cell proliferation in the TBI animals, leading to preserved cognition. Figure constructed via Biorender.com.

## Data Availability

Not applicable.

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
