# Peer review of "Stem Cell Therapy for Sequestration of Traumatic Brain Injury-Induced Inflammation"

_ijms, 2022, doi:10.3390/ijms231810286_

Round 1
Reviewer 1 Report
In this manuscript, the authors described the inflamed brain after traumatic brain injury (TBI) and radiation injury; they reviewed the status of stem cells in the inflamed brain and the applications of cell therapy in sequestering inflammation in TBI.
This manuscript is too short and unfortunately needs very substantial improvements and corrections before publishing may be possible.
General points:
Please add a list of abbreviations before References section to your manuscript.
Special points:
For better readability, please add at least two additional Figures to this review.
For better readability and importance for this review, please describe the TBI more exactly and add to your manuscript the exactly description of following points:
Incidence of TBI
Mechanisms of TBI
Age-specific incidence of TBI
Gender difference of TBI
Severity of TBI
Symptoms
Mortality of TBI
Treatment
Prognosis
Alzheimer’s disease and TBI
Please add to your manuscript the “Future perspectives” section.
Also important, this manuscript should be substantially improved, i. e., by substantial references in the field.
Key words: please add also to keywords: stem cell-based therapy; traumatic brain injury
Main part of the manuscript:
Lines 29-31: please add multiple references at the end of this sentence.
Lines 41-45: please describe all these publication very exactly.
Lines 49-51: please add multiple references at the end of this sentence.
Lines 62-64: please add multiple references at the end of this sentence.
Lines 101-105: please add multiple references at the end of each these sentences.
Lines 108-109: please add multiple references at the end of this sentence.
Author Response
Reviewer 1:
In this manuscript, the authors described the inflamed brain after traumatic brain injury (TBI) and radiation injury; they reviewed the status of stem cells in the inflamed brain and the applications of cell therapy in sequestering inflammation in TBI. This manuscript is too short and unfortunately needs very substantial improvements and corrections before publishing may be possible. Response: We appreciate the Reviewer’s comments and have now made substantial improvements to the manuscript incorporating all the suggestions as detailed below.
General points:
Please add a list of abbreviations before References section to your manuscript. Response: We have added an abbreviations section.
Special points:
For better readability, please add at least two additional Figures to this review. Response: We have added two additional figures, one showing inflammation as likely the merging pathway for both TBI and radiation-induced injury and another how stem cells may sequester inflammation-induced secondary cell death via an anti-inflammatory mechanism, i.e., secretion of anti-inflammatory factors.
For better readability and importance for this review, please describe the TBI more exactly and add to your manuscript the exactly description of following points:
Incidence of TBI
Mechanisms of TBI
Age-specific incidence of TBI
Gender difference of TBI
Severity of TBI
Symptoms
Mortality of TBI
Treatment
Prognosis
Alzheimer’s disease and TBI
Response: We have now added all the topics above to our paper – please see section 1.
Please add to your manuscript the “Future perspectives” section. Response: We have added a Future Perspective section.
Also important, this manuscript should be substantially improved, i. e., by substantial references in the field. Response: We have added references (45 new citations) to support our thesis.
Key words: please add also to keywords: stem cell-based therapy; traumatic brain injury. Response: We have added the suggested keywords.
Main part of the manuscript:
Lines 29-31: please add multiple references at the end of this sentence.
Lines 41-45: please describe all these publicatios very exactly.
Lines 49-51: please add multiple references at the end of this sentence.
Lines 62-64: please add multiple references at the end of this sentence.
Lines 101-105: please add multiple references at the end of each these sentences.
Lines 108-109: please add multiple references at the end of this sentence.
Response: All the suggestions above have been incorporated to our paper.
Reviewer 2 Report
In this concise review, Borlongon and Rosi describe how TBI elicit the mobilization of peripheral immune cells to influence the inflammatory sequelae in the CNS. Although the premise of the manuscript covers an important component of immunity following acute brain injury, the authors did not adequately describe how neurogenic signals from the injured brain alters the bone marrow niche or the hematopoietic response to such injuries. It has been experimentally demonstrated in other acute brain injury paradigms of sterile inflammation, mainly in ischemic and hemorrhagic strokes, of acute changes to hematopoietic stem cells and their subsequent bias towards myelopoiesis and resulting consequences on neuroinflammation. Authors should reference these works in major journals and elucidate similar mechanisms or neurogenic insights, if similar papers are not reported in TBI. This would be more informative and scientifically relevant than their extension towards tangential findings in space flight and irradiation.
Author Response
Reviewer 2:
In this concise review, Borlongan and Rosi describe how TBI elicit the mobilization of peripheral immune cells to influence the inflammatory sequelae in the CNS. Although the premise of the manuscript covers an important component of immunity following acute brain injury, the authors did not adequately describe how neurogenic signals from the injured brain alters the bone marrow niche or the hematopoietic response to such injuries. It has been experimentally demonstrated in other acute brain injury paradigms of sterile inflammation, mainly in ischemic and hemorrhagic strokes, of acute changes to hematopoietic stem cells and their subsequent bias towards myelopoiesis and resulting consequences on neuroinflammation. Authors should reference these works in major journals and elucidate similar mechanisms or neurogenic insights, if similar papers are not reported in TBI. This would be more informative and scientifically relevant than their extension towards tangential findings in space flight and irradiation. Response: We thank the Reviewer for the valuable comments and have now a more in-depth discussion the potential neurogenic signals released by the injured brain that may potentially affect the response of the bone marrow niche or the hematopoietic to neural injuries. To this end, we discuss how the process of myelopoiesis and the subsequent inflammation in response to injury may propel innate immune cells, such as neutrophils, dendritic cells and monocytes, develop from a myeloid progenitor cells to the site of injury. We refer to ischemic and hemorrhagic stroke, as well as TBI in building this concept of injury-induced myelopoiesis. Relevant references to this topic are provided. We then present some scenarios on how myelopoiesis and inflammation observed in stroke and TBI may similarly apply to space flight and irradiation, which is the novel topic introduced here since they have never been covered in the reported literature.
Round 2
Reviewer 1 Report
Thank you for all corrections.